# Is There a Place for Elastography in the Diagnosis of Hepatocellular Carcinoma?

**DOI:** 10.3390/jcm10081710

**Published:** 2021-04-15

**Authors:** Ana-Maria Ghiuchici, Ioan Sporea, Mirela Dănilă, Roxana Șirli, Tudor Moga, Felix Bende, Alina Popescu

**Affiliations:** Department of Gastroenterology and Hepatology, “Victor Babeș” University of Medicine and Pharmacy, 300041 Timișoara, Romania; ghiuchici.anamaria@umft.ro (A.-M.G.); isporea@umft.ro (I.S.); sirli.roxana@umft.ro (R.Ș.); moga.tudor@umft.ro (T.M.); bende.felix@umft.ro (F.B.); popescu.alina@umft.ro (A.P.)

**Keywords:** hepatocellular carcinoma, elastography, virtual touch quantification, liver cirrhosis

## Abstract

Background and Aims: Elastography can provide information regarding tissue stiffness (TS). This study aimed to analyze the elastographic features of hepatocellular carcinoma (HCC) and the factors that influence intratumoral elastographic variability in patients with liver cirrhosis. Methods: This prospective study included 115 patients with liver cirrhosis and hepatocellular carcinoma evaluated between June 2016–November 2019. A total of 88 HCC nodules visualized in conventional abdominal ultrasound (US) met the inclusion criteria and underwent elastographic evaluation. Elastographic measurements (EM) were performed in HCC and liver parenchyma using VTQ (Virtual Touch Quantification), a point shear wave elastography (pSWE) technique. In all patients, we performed contrast-enhanced ultrasound (CEUS), and the final diagnosis of HCC was established by contrast-enhanced-CT or contrast-enhanced-MRI. Results: The mean VTQ values in HCCs were 2.16 ± 0.75 m/s. TS was significantly lower in HCCs than in the surrounding liver parenchyma 2.16 ± 0.75 m/s vs. 2.78 ± 0.92 (*p* < 0.001). We did not find significant differences between the first five and the last five EM, and the intra-observer reproducibility was excellent ICC: 0.902 (95% CI: 0.87–0.950). However, the tumor size, heterogeneity, and depth correlated with higher intralesional stiffness variability (*p* < 0.001). Conclusions: VTQ brings additional information for HCC characterization. Intra-observer reproducibility for both HCC and liver parenchyma was excellent. Knowing the stiffness of HCC’s might endorse an algorithm-based approach towards focal liver lesions (FLLs) in liver cirrhosis.

## 1. Introduction

Bi-annual ultrasound (US) surveillance is recommended worldwide as a screening tool for HCC in patients with cirrhosis by all major hepatology societies [1,2,3,4]. Conventional US is a widely available, non-invasive, risk-free, inexpensive imaging tool that provides a real-time assessment of the liver. Having all these advantages, US is the most commonly used tool in the diagnostic algorithm of liver disease. Greyscale (B-mode) imaging and Doppler US are available on almost all US machines, but newer devices have the additional option of elastography and contrast-enhanced ultrasound (CEUS), providing more information, a faster and more accurate diagnosis, usually at the point of care. This approach shortens the diagnostic algorithm duration and can avoid further long-waiting procedures such as CT/MRI and/or liver biopsy.

Liver elastography is widely used in clinical practice for liver stiffness assessment in patients with chronic liver disease as a non-invasive marker of fibrosis. Other applications of shear wave elastography in the field of hepatology include diagnosing clinically significant portal hypertension (CSPH) and high-risk oesophageal varices (HRV), characterization of FLLs, and prognosis of clinical outcomes in chronic liver disease [5,6,7,8]. The US elastography techniques have as fundamental principle measurement of minimal displacements in the tissue caused by mechanical compression or by an enforced acoustic impulse which acts as a wavefront [9]. These displacements lead to the formation of shear waves in the tissue, which spread faster in a stiff medium (i.e., cirrhotic liver).

FLLs are commonly first detected by US, and, for a definite diagnosis, contrast-enhanced imaging and/or biopsy are needed. FLLs show different stiffness due to their different histological structure. Different tissue stiffness ratio values can be observed when comparing FLL stiffness with adjacent liver parenchyma due to various underlying pathology. Thus, performing elastography during US evaluation can be a helpful tool that can assess elasticity proprieties and provide information for the differential diagnosis of various FLLs and finally, if used in a multiparametric US approach, can help to avoid biopsy (an invasive procedure) and resorting to CE-CT/CE-MRI (radiation exposure and potentially nephrotoxic contrast agents, expensive, difficult re-attendance).

Several studies showed that elastography could bring additional information regarding FLL stiffness and might even predict their nature. However, a significant amount of heterogeneity was found in the studies published to date [10,11,12,13,14,15,16]. Despite promising results, currently, the EFSUMB guidelines do not recommend elastography to evaluate the stiffness of FLLs to differentiate between benign and malignant lesions in clinical practice, but remains a field of further research [9]. Since these methods are currently undergoing validation, elastographic features of different FLLs and clinical contexts should be closely evaluated.

In this paper, we aimed to analyze the elastographic features of HCC and the factors that influence intratumoral elastographic variability in patients with liver cirrhosis.

## 2. Materials and Methods

### 2.1. Study Design and Population Selection

This prospective study included consecutive patients with liver cirrhosis diagnosed with HCC during US surveillance in a tertiary Gastroenterology and Hepatology Department.

From 115 cirrhotic patients with 121 focal liver lesions found during US surveillance for HCC evaluated between June 2016–November 2019, 88 lesions were included in the final study cohort. The inclusion criteria were: FLLs ≥ 2 cm, adequate visualization of the FLLs using conventional US, at a maximum depth of 8 cm from the skin surface, and a final diagnosis of HCC established by CE-CT or CE-MRI (typical enhancement patterns for HCC during contrast imaging assessment) in a suggestive clinical and biologic context. The exclusion criteria were: FLL < 2 cm, depth over 8 cm from the skin surface, large perihepatic ascites, poor compliance of the patient (inability to hold their breath during the examination), and lesions in which “x.xx” (not applicable) results were obtained. This prospective study did not evaluate any patient with benign or non-HCC FLLs. We established the diagnosis of cirrhosis using clinical, biologic, US, and endoscopic criteria.

### 2.2. VTQ Evaluation

EM were performed inside the FLLs and in the surrounding liver parenchyma using VTQ, a pSWE method using acoustic radiation force impulse technique (ARFI). VTQ was performed with the Siemens Acuson S2000^TM^ ultrasound system (Siemens AG, Erlangen, Germany) using a curved array probe. All VTQ measures of FLLs and surrounding liver were performed by a single ultrasound expert operator with more than two years of experience in pSWE elastography.

The patients were examined in a supine or left/right lateral decubitus posture depending on the FLLs localization. After the lesion was identified by conventional US imaging, the VTQ measurements were performed by placing the region of interest (ROI) box inside the analyzed tumor. The ROI box had a fixed size of 10 × 5 mm and was placed in the solid portion of the lesion, avoiding vascular or necrotic tissue. Measurements were performed in intermediate breath-hold in order to avoid motion artifacts. We performed ten EM in the FLL and ten EM in the surrounding liver parenchyma, at approximately the same depth, 2–3 cm away from the lesion. In the case of multiple lesions, the largest FLL or the FLL best visualized by US was considered for evaluation. In the case of larger FLLs, the ROI was positioned at different points to obtain EM.

The VTQ results were expressed in m/s as the mean value of the 10 measurements. We assessed the intra-observer reproducibility and compared the first 5 and the last 5 EM. The ratio of VTQ measurements of each FLL vs. the surrounding liver parenchyma was also calculated. Figure 1 shows a VTQ measurement in an HCC nodule.

According to the data provided by Siemens Corporation, the SWV range from 0.5 (softer or cystic portions of FLLs) to 5.00 m/s (harder or calcific portions of FLLs); any value outside of this range is displayed as ‘‘x.xx m/s’’, which means not applicable (NA).

### 2.3. CEUS Evaluation

CEUS examinations comply with EFSUMB guidelines for the characterization of focal liver lesions [17]. Physicians with high expertise in hepatobiliary ultrasound and CEUS (level III Experts according to the EFSUMB classification) performed the CEUS examinations. All contrast studies were performed using SonoVue^®^ (Bracco Spa, Milan Italy). The lesion enhancement pattern was assessed and documented. An independent operator performed VTQ EM in HCC and surrounding liver parenchyma. Previous to EM, the elastography operator reviewed all CEUS video clips in order to avoid performing EM in the necrotic areas of the tumor.

The study protocol was approved by the local Ethics Committee, being in accordance with the Helsinki Declaration of 1975, and all subjects agreed to undergo EM and imaging assessment as part of their medical workup.

### 2.4. Statistical Analysis

The statistical analysis was performed using IBM SPPS^®^ statistics for Windows, V 20.0 (Armonk, NY, USA: IBM Corp) and Microsoft Office Excel 2010. Baseline characteristics of the FLLs were shown according to their origin using descriptive statistics. Quantitative variables are expressed as a mean ± standard deviation, absolute numbers, and percentages. We assessed the intra-observer reproducibility of VTQ by using interclass correlation coefficients (ICCs) with 95% lower and upper limits of agreement (LOA). ICC values were interpreted as follows: poor (ICC 0–0.20), fair (0.21–0.40), good (ICC 0.41–0.75) and excellent (ICC ≥ 0.75). The Friedman test was used to compare the first five and last five EMs taken by the elastographic operator.

## 3. Results

### 3.1. Baseline Characteristics

The final study group included 88 lesions from 88 patients with liver cirrhosis. The success rate for VTQ in HCC was 72.7% (88/121). Patients’ characteristics are summarized in Table 1.

### 3.2. VTQ Values

The mean VTQ values inside the HCCs were 2.16 ± 0.75 m/s. TS was significantly lower in HCCs than in the surrounding liver parenchyma 2.16 ± 0.75 m/s vs. 2.78 ± 0.92 (*p* < 0.001). Nodules’ characteristics are shown in Table 2.

VTQ mean nodules’ size for all HCCs (*n* = 88) in our studied group was 4.9 ± 2.2 cm. When we considered the nodules’ size, we obtained a higher intra-tumoral variability of EM values in the larger nodules *p* < 0.001. We selected a cut-off value of 3 cm having in mind the early diagnosis in small HCCs that could be suitable for percutaneous treatment. We calculated VTQ means separately, according to a threshold diameter of 3 cm (Table 3).

### 3.3. CEUS Examination

CEUS was considered conclusive for HCC if the typical enhancement pattern was present: hyperenhancement in the arterial vascular phase with late-onset (>60 s) washout of mild intensity. The typical enhancement pattern and a conclusive diagnosis of HCC were obtained in 76.1% of cases (67/88) as compared with the reference method (contrast-enhanced CT/MRI). Comparing the means of EM in HCC with conclusive CEUS vs. inconclusive CEUS, no significant difference was observed 2.12 ± 0.58 vs. 2.10 ±0.62 (*p* < 0.001).

### 3.4. Intra-Observer Reproducibility

We tested the intra-observer reproducibility for VTQ in tumoral and surrounding liver parenchyma stiffness assessment.

The intra-observer reproducibility for EM in liver parenchyma was excellent ICC: 0.964 (95% CI: 0.943–0.960).

The intra-observer reproducibility for EM in HCC was excellent ICC: 0.902 (95% CI: 0.87–0.950), proving the method to be reliable also for tumor evaluation.

The good ICCs for EM values indicate that VTQ is a reproducible method in assessing both tumoral and liver stiffness. 

The ICC between the medians of the first five and last five EM was high and statistically significant (ICC: 0.926 (95% CI: 0.890–0.960). No significant differences (*p* = 0.75) were found when comparing the first five and the last five EM, suggesting that in practice, five measurements in the tumor and five measurements in the liver tissue could be enough for evaluation.

## 4. Discussion

Liver cirrhosis is the main risk factor for HCC development. Therefore practice guidelines recommend US surveillance for early detection of HCC in this category of patients [1,2,3,4]. HCC can be diagnosed non-invasively by contrast-enhanced imaging (CE-CT; CE-MRI; CEUS) if a typical enhancement pattern is present [2,17]. Considering that well-differentiated HCCs often lack arterial hyper-enhancement, appearing iso- or even hypoenhanced in the arterial phase, and some well-differentiated HCC do not show washout at all [17], HCC diagnosis can be challenging by imaging methods.

CEUS has good performance for HCC diagnosis comparable to CE-CT and CE-MRI [18,19]. The size of the nodule can modify CEUS sensitivity. SRUMB study [19] showed a lower sensitivity of CEUS in small HCCs ≤ 2 cm as compared to HCC > 2 cm, 56.3% vs. 78.9%. In our study, CEUS examination was conclusive for HCC diagnosis in 76.1% of cases (all examined nodules were ≥2 cm).

The correct characterization of a nodule encountered in a cirrhotic liver is important for further management (follow-up/suitable therapeutic option). Liver biopsy is limited in patients with liver cirrhosis due to possible complications but should be considered when imaging techniques cannot establish a confident diagnosis [2,20].

US evaluation might compete with other advanced imaging techniques similar to contrast-enhanced CT or MRI with the new features implemented in US machines. The new ultrasound features, besides the greyscale imaging, Doppler, and color Doppler mode, can perform multiple elastographic methods that enable us to assess the fibrosis, and with the introduction of contrast medium (contrast-enhanced ultrasound or CEUS), the paradigm of ultrasound indication has changed thus the multiparametric US concept appeared offering a broader perspective of the examined structures [21,22]. By providing contrast to US, a broad spectrum of features have been implemented into the US machines allowing us to quantitatively analyze the organ perfusion in terms of time and intensity (TIC analysis). 3D/4D US fusion techniques are also applications used for real-time reconstruction of examined structures finding their utility in some medical fields, even for difficult liver lesions demanding percutaneous or surgical interventions [23].

Several proposed practical algorithms for the clinical use of MPUS in chronic liver disease and FLL are available [24], and computer-aided diagnosis systems were conceived based on the new US features [25,26].

Previous studies demonstrated that malignant FLLs are generally stiffer than benign lesions, reporting the following descending stiffness order: Liver metastases > HCC > FNH (focal nodular hyperplasia) > Hemangioma [10,27,28]. In the setting of liver cirrhosis, HCC lesions may appear softer than the surrounding liver parenchyma and also softer than other malignant FLLs (metastases and cholangiocarcinoma) [28]. Similar to our results, Gallotti et al. [29] showed HCCs are softer lesions compared to the surrounding liver parenchyma with a mean shear-wave velocity in HCC of 2.17 vs. 2.99 m/s.

Grgurevic et al. [30] developed a liver elastography malignancy prediction score (LEMP) for non-invasive characterization of focal liver lesions that enables correct differentiation of benign and malignant FLL in 96% of patients. The authors concluded that RT-2D-SWE (real-time 2-dimensional share-wave elastography) could be a reliable method for differentiating malignant from benign liver lesions with a comprehensive approach.

Clinical context influences the diagnostic performance of elastography [28]. In clinical practice, the differential diagnosis of FLLs requires analysis of various data, including age, underlying liver disease, serum biomarkers, and findings with other imaging modalities. Our study focuses on the elastographic features of HCC in cirrhotic patients that have a fibrotic background liver. VTQ values in HCCs were 2.16 ± 0.75 m/s, significantly lower compared to the surrounding liver parenchyma stiffness with VTQ ratio of 1.33 ± 0.66 m/s, showing HCC as a softer tissue as compared to the stiff parenchyma of the cirrhotic liver.

A strength of this study is the homogeneity of the included lesions, all HCCs, given the difficulties in diagnosing these FLLs in clinical practice. It was also possible to integrate elastography with US end CEUS to obtain a multiparametric US approach easily applicable in daily practice in all analyzed lesions. Knowing the stiffness features of HCC could be helpful in clinical practice if we have an inconclusive result for the CEUS exam and a mean value of pSWE showing a soft FLL compared to the surrounding parenchyma, having in mind the results of this study, HCC suspicion could be raised. Elastography can be a useful tool in orienting the diagnosis and the need for rapid further assessment. VTQ can easily and inexpensively be integrated into imaging protocols already involving standard US and CEUS.

The excellent ICCs for the mean values show that VTQ pSWE for evaluating FLL stiffness is a reproducible method and could provide complementary information regarding the TS, useful for the differential diagnosis of FLLs, if properly interpreted. The intra-observer reproducibility for EM in HCC was excellent ICC: 0.902 (95% CI: 0.87–0.950). Bota et al. showed in a study regarding ARFI reproducibility an excellent overall intra-operator agreement (ICC 0.90) [31]. We did not find significant differences between the first five and the last five EM showing that 5 EM are enough for obtaining reliable results.

Even if our study included a relatively large number of HCCs, it also has limitations: we did not include other benign/malignant liver lesions found in cirrhotic patients; no biopsies for the analyzed FLLs were available. A limitation for the use in clinical practice is the fact that it is a time-consuming procedure that took up to 20–30 min in some cases. Regarding the place of elastography for HCC diagnosis, further studies are still required to obtain an evidence-based answer for the question raised in this paper’s title.

## 5. Conclusions

VTQ brings additional information for HCC characterization regarding tumoral stiffness. The good ICCs for EM values show that VTQ is a reproducible method in assessing both tumoral and liver stiffness.

Knowing the stiffness of HCC’s might endorse an algorithm-based approach towards FLL’s in liver cirrhosis.

## Figures and Tables

**Figure 1 jcm-10-01710-f001:**
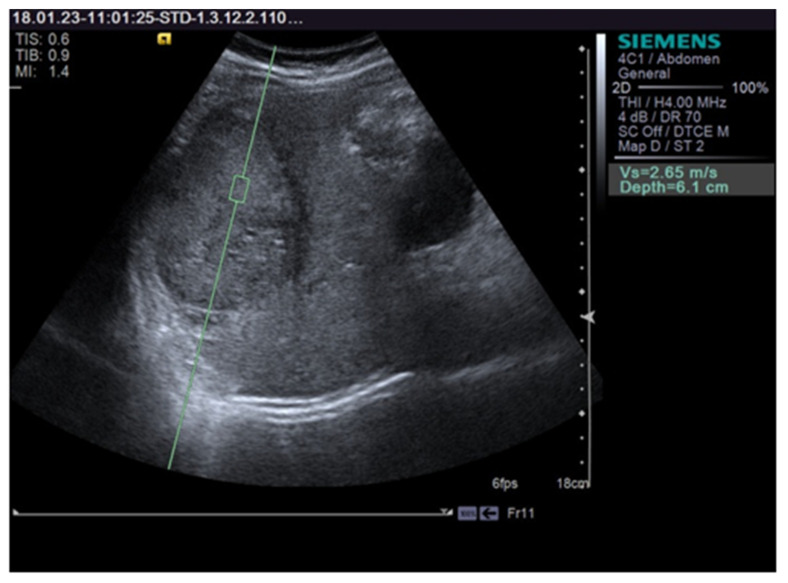
Shows a VTQ (Virtual Touch Quantification) measurement in an hepatocellular carcinoma (HCC) nodule.

**Table 1 jcm-10-01710-t001:** Patients’ characteristics.

Variable	Patients (*n* = 88)
Age (years)	62 ± 9.5
Male	56 (63.3)
Underlying disease	
Compensated cirrhosis	71 (80.7)
Decompensated cirrhosis	17 (19.3)
Etiologies of cirrhosis	
HCV	38 (43.2)
Alcohol	22 (25)
HBV	23 (26.1)
Multiple etiologies	5 (5.7)

The results are expressed as mean ± SD (range) or number (%). HCV; hepatitis C virus, HBV; hepatitis B virus.

**Table 2 jcm-10-01710-t002:** Nodules’ characteristics.

Variable	HCC Nodules (*n* = 88)
Size (cm)	4.9 ± 2.2
Depth (cm)	5.1 ± 1.8
VTQ mean in HCC (m/s)	2.16 ± 0.75
VTQ mean in liver parenchyma (m/s)	2.78 ± 0.92
VTQ ratio (m/s)	1.33 ± 0.66

The results are expressed as mean ± SD (range). HCC; hepatocellular carcinoma, VTQ; virtual touch quantification.

**Table 3 jcm-10-01710-t003:** VTQ means (m/s) for HCCs with a 3 cm threshold.

HCC (*n* = 88)	VTQ Mean (m/s)
HCC ≤ 3 cm (*n* = 24)	2.05 ± 0.67
HCC > 3 cm (*n* = 64)	2.21 ± 0.78

The results are expressed as mean ± SD (range). HCC; hepatocellular carcinoma, VTQ; virtual touch quantification.

## Data Availability

Due to ethical reasons, the data are not publicly available. All anonymized data presented in this study are available on request from the corresponding author.

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
