# Peer review of "Is There a Place for Elastography in the Diagnosis of Hepatocellular Carcinoma?"

_jcm, 2021, doi:10.3390/jcm10081710_

Round 1
Reviewer 1 Report
General comments:
This prospective study evaluates the elastography measurements in HCC and liver parenchyma in 88 patients. The main study results are the significantly lower stiffness of HCC compared to background cirrhotic liver parenchyma and the excellent inter-reader agreement for elastography measurements. Overall, the lack of a control group significantly limits the clinical impact of the provided results, and it is unclear how this could improve HCC characterization based on the difference between HCC and background parenchyma.
Specific comments:
Abstract:
-Methods: This study included 115 patients but only 88 HCCs. This mean that not all the patients had HCC. Why the Authors included also non-HCC patients?
-Conclusion: “VTQ brings additional information for HCC characterization”. It is difficult to prove this conclusion since no control group was assessed and the Authors did not compare with histopathological parameters. What information add this study compared to the HCC characterization on contrast-enhanced imaging?
Introduction:
-Page 2, lines 52-53: “FLLs also have different stiffness as compared to the surrounding tissue due to their different histological structure”. How this data could be useful to avoid biopsy or other imaging in clinical practice? The more relevant clinical question could be if stiffness could differentiate benign from malignant FLLs.
Methods:
-CEUS evaluation: Please specify the type of administered contrast agent for CEUS examinations.
-CEUS evaluation, lines 112-113: “Previous to EM, the elastography operator reviewed all CEUS videoclips in order to avoid performing EM in the necrotic areas of the tumor”. It looks that all patients underwent CEUS before VTQ evaluation. Were the two examination performed by different operators? From this sentence it looks that the elastography operator was not blinded to the FLLs diagnosis.
Results:
-Lines 131-133: “27 patients were excluded because of the tumor size and depth (FLL smaller than 2 cm or deeper 132 than 8 cm), large perihepatic ascites..”. It is suggested to move these exclusion criteria in the methods. This will help to better understand who the final population was derived. For instance, this prospective population did not evaluate any patient with benign or non-HCC FLLs.
-Lines 162-163: “When we considered the nodules’ size, we obtained a higher variability of EM values in the larger nodules”. Was this comparison statistically significant? How was selected the cutoff value of 3 cm?
-Lines 169-170: “We did not find significant differences between the first five and the last five EM (p<.750)”. Should it be p>.750? P-value<0.750 is generic and include statistically significance level.
-Intra-observer reproducibility: Did the Authors evaluated also intra-observer reproducibility of EM in liver parenchyma? Consider to report this data if available.
-CEUS examination is mentioned in the methodology, but no results are provided on CEUS assessment. It is suggested to report CEUS assessment results if relevant or delete from methods, reporting CEUS examination in aggregate with CT and MRI assessment.
Discussion:
-Lines 206-209: “In the setting of liver cirrhosis, HCC lesions may appear softer than the surrounding liver parenchyma and also softer than other malignant FLLs”. The lower stiffness of HCC compared to the back ground parenchyma is expected in cirrhotic patients due to the elevate liver stiffness. How this could help in the characterization of HCC? What are the impact in clinical practice for HCC characterization of the provided results?
-Consider to discuss the possible advances in knowledge of this study compared to prior evidences.
-Conclusion: “VTQ brings additional information for HCC characterization”. Again it is very difficult to state that VTQ provides additional information since there is no comparison with other lesions of enhancement pattern of HCC. How the difference between HCC and liver parenchyma could be an additional information for characterization of HCC? I suggest to report in the conclusion the excellent agreement for stiffness measurements.
Author Response
Dear Editor and Reviewers,
Thank you very much for your comments and suggestions for our manuscript JCM-1148663, entitled "Is there a place for elastograhpy in the diagnosis of hepatocellular carcinoma?", which are very valuable and have helped us revise and improve our manuscript. We appreciate all your work on our manuscript. We have carefully studied your comments and corrected our manuscript accordingly. We hope this version of the manuscript meets your approval. All changes have been highlighted in the manuscript. The corrections in the paper and the response to the editor and reviewers are as follows:
Response to Reviewer 1 Comments
Point 1: Abstract/Methods: This study included 115 patients but only 88 HCCs. This mean that not all the patients had HCC. Why the Authors included also non-HCC patients?
Response 1: Thank you for your comment. We agree that this statement can be confusing for the reader. We included in this study only patients with liver cirrhosis and HCC, but only 88 HCCs met the inclusion criteria; in 27 patients, we could not assess HCC stiffness due to several limitations (tumor size < 2cm, depth, large perihepatic ascites).
page1; line15.
Point 2: Abstract/Conclusion: “VTQ brings additional information for HCC characterization”. It is difficult to prove this conclusion since no control group was assessed and the Authors did not compare with histopathological parameters. What information add this study compared to the HCC characterization on contrast-enhanced imaging?
Response 2: Thank you for your comment. A group control represents a future study goal for us, and we agree that the absence is a significant limitation for this paper. We had only a few cases of benign lesions in cirrhotic patients and no histology exams. That was the main reason for not including those cases in the study. Keeping in mind that most studies that analyzed FLL stiffness had inhomogeneous groups, and the significant overlap between SWE values was observed that limits the use of this technique in clinical practice, we aimed to study the elastography features of a single type of FLL knowing the clinical background. Our main objective was to assess HCC stiffness with definite imagistic diagnostic in a homogenous group (cirrhotic patients). We added information about CEUS examination and HCC stiffness in the results section. “VTQ brings additional information for HCC characterization”- the additional information is the FLL and ratio stiffness. Knowing the stiffness features of HCC could be helpful in clinical practice. In cirrhotic patients, when an FLL seen in the standard US is further analyzed by CEUS. If we have an inconclusive result for the CEUS exam but a mean value of SWE showing a soft FLL compared to the surrounding parenchyma, we could think of HCC having in mind the results of this study.
page 5; line194-201.
Point 3: Introduction/Page 2, lines 52-53: “FLLs also have different stiffness as compared to the surrounding tissue due to their different histological structure”. How this data could be useful to avoid biopsy or other imaging in clinical practice? The more relevant clinical question could be if stiffness could differentiate benign from malignant FLLs.
Response 3: Thank you for your comment. Yes, tumor differentiation is essential, but given that a cut-off value has not been established so far (the cut-off value of SWE is different across studies), we believe that several factors need to be considered to see what causes this variability. Since these methods are currently undergoing validation, elastography features of different FLLs and clinical contexts should be closely evaluated. We believe that elastograhpy is not enough for tumor differentiation but could be helpful in clinical practice. VTQ can easily and inexpensively be integrated into imaging protocols already involving the use of standard US and CEUS.
page2; line58-59; 68-70.
Point 4: Methods/CEUS evaluation: Please specify the type of administered contrast agent for CEUS examinations.
Response 4: Thank you for your comment. All contrast studies were performed using SonoVue® (Bracco Spa, Milan, Italy) as a contrast agent. We have added this information to the text.
page3; line 126-127.
Point 5: Methods/ CEUS evaluation, lines 112-113: “Previous to EM, the elastography operator reviewed all CEUS videoclips in order to avoid performing EM in the necrotic areas of the tumor”. It looks that all patients underwent CEUS before VTQ evaluation. Were the two examination performed by different operators? From this sentence it looks that the elastography operator was not blinded to the FLLs diagnosis.
Response 5: Thank you for your comment. CEUS exams were performed by physicians with high expertise in hepatobiliary ultrasound and CEUS (level III Experts according to the EFSUMB classification), a different operator performed elastography measurements. Yes, the elastography operator was not blinded to the FLLs diagnosis because in this study we did not analyse the potential of ultrasound based elastography for differentiating benign from malignant FLL.
page3; line 124-128.
Point 6: Results/Lines 131-133: “27 patients were excluded because of the tumor size and depth (FLL smaller than 2 cm or deeper 132 than 8 cm), large perihepatic ascites.”. It is suggested to move these exclusion criteria in the methods. This will help to better understand who the final population was derived. For instance, this prospective population did not evaluate any patient with benign or non-HCC FLLs.
Response 6: Thank you for your suggestion. We made the corrections accordingly.
page2; line 85-88.
Point 7: Results/Lines 162-163: “When we considered the nodules’ size, we obtained a higher variability of EM values in the larger nodules”. Was this comparison statistically significant? How was selected the cut-off value of 3 cm?
Response 7: Thank you for your comment. We selected the cut-off value having in mind the early diagnosis in small HCC that could be suitable for percutaneous treatment. Higher variability of EM was observed in large lesions > 3 cm (p<.001).
page4, line176-178.
Point 8: Results/ Lines 169-170: “We did not find significant differences between the first five and the last five EM (p<.750)”. Should it be p>.750? P-value<0.750 is generic and include statistically significance level.
Response 8: Thank you for your comment. We included the statistically significance level.
page5; line 203-204.
Point 9: Results/Intra-observer reproducibility: Did the Authors evaluated also intra-observer reproducibility of EM in liver parenchyma? Consider to report this data if available.
Response 9: Thank you for your comment. We assessed also the intra-observer reproducibility in liver parenchyma that was excellent. We reported the results in the manuscript.
page; line208-209; 212-213
Point 10: Results/ CEUS examination is mentioned in the methodology, but no results are provided on CEUS assessment. It is suggested to report CEUS assessment results if relevant or delete from methods, reporting CEUS examination in aggregate with CT and MRI assessment.
Response 10: Thank you for your comment. We added CEUS examination results. (We obtained a conclusive diagnostic for HCC in 76.1% cases; the EM in the cases with conclusive CEUS for HCC were not significantly different compared with the cases of inconclusive CEUS).
page5; line194-201.
Point 11: Discussion/Lines 206-209: “In the setting of liver cirrhosis, HCC lesions may appear softer than the surrounding liver parenchyma and also softer than other malignant FLLs”. The lower stiffness of HCC compared to the background parenchyma is expected in cirrhotic patients due to the elevate liver stiffness. How this could help in the characterization of HCC? What are the impact in clinical practice for HCC characterization of the provided results?
Response 11: Thank you for your comment. In cirrhotic patients, if we have an inconclusive result for the CEUS exam of an FLL but a mean value of SWE showing a soft FLL compared to the surrounding parenchyma, HCC could be considered as a final diagnosis. Thus, by integrating elastography with US and CEUS in a multiparametric US approach easily applicable in practice, we can improve our diagnosis.
page 6; line 272-280.
Point 12: Discussion/ Consider to discuss the possible advances in knowledge of this study compared to prior evidences.
Response 12: Thank you for your comment. Even though we could not assess this method's potential for tumoral differentiation, a strength point of this study is the homogeneity of the included lesions, all HCCs, given the difficulties in diagnosing these FLL in clinical practice. Also, it was possible to integrate elastography with US and CEUS to obtain a multiparametric US approach easily applicable in daily practice in all analyzed lesions.
page 6; line 272-280.
Point 13: Conclusion: “VTQ brings additional information for HCC characterization”. Again it is very difficult to state that VTQ provides additional information since there is no comparison with other lesions of enhancement pattern of HCC. How the difference between HCC and liver parenchyma could be an additional information for characterization of HCC? I suggest to report in the conclusion the excellent agreement for stiffness measurements.
Response 13: Thank you for your comment. As we previously emphasized, we consider that knowing the stiffness features of HCC could be helpful in clinical practice. In cirrhotic patients, if we have an inconclusive result for the CEUS exam of a FLL but a mean value of SWE showing a soft FLL compared to the surrounding parenchyma, having in mind the results of this study we could think of HCC. We agree with the reviewer and we add in the conclusion also the excellent agreement for stiffness measurements.
page 6; line 272-280; 296-297.
Reviewer 2 Report
The authors report finding significantly lower SWE readings for HCC in comparison to background cirrhotic livers. This could be helpful to improve the sensitivity of U/S in making HCC diagnosis.
The idea is not novel and has been extensively studied and analysed in the past. Overall, the question- Is there a place for elastography in the diagnosis of hepatocellular carcinoma (HCC)?- is not answered by the study.
1) The authors need to emphasize on the novel aspects of their study. Authors confirm a good intra-observer reproducibility, however, it is already well known for other studies on SWE.
2) the key question is whether the technique distinguishes cancer from regenerative nodule. But, the study has just characterised HCC.
3) The application may be limited in clinical practice as it is only useful once a focal lesion is identified on ultrasound to enable targeted measurement of SWE. However, the main problem with ultrasound is the fact that focal lesions are not seen/identified in the first place. If lesions are not recognized/identified on US, targeted SWE will not get measured in clinical practice.
4) Another limitation is that the study does not highlight how a low SWE reading would discriminate HCC from other hepatic focal lesion differentials.
5) The authors report finding significantly lower SWE readings for HCC in comparison to background cirrhotic livers. This could be helpful to improve the sensitivity of U/S in making HCC diagnosis. But, this hasn't been tested in this study.
Author Response
Dear Editor and Reviewers,
Thank you very much for your comments and suggestions for our manuscript JCM-1148663, entitled "Is there a place for elastograhpy in the diagnosis of hepatocellular carcinoma?", which are very valuable and have helped us revise and improve our manuscript. We appreciate all your work on our manuscript. We have carefully studied your comments and corrected our manuscript accordingly. We hope this version of the manuscript meets your approval. All changes have been highlighted in the manuscript. The corrections in the paper and the response to the editor and reviewers are as follows:
Response to Reviewer 2 Comments
Point 1: The authors need to emphasize on the novel aspects of their study. Authors confirm a good intra-observer reproducibility, however, it is already well known for other studies on SWE.
Response 1: Thank you for your comment and for your suggestion.
Most studies that analysed FLL stiffness had inhomogeneous groups, and significant overlap between SWE values was observed that limits the use of this technique in clinical practice. We aimed to study the elastography features of a single type of FLL knowing the clinical background. Our main objective was to assess HCC stiffness with definite imagistic diagnostic in a homogenous group (cirrhotic patients).
A strength point of this study is the homogeneity of the included lesions, all HCCs, given the difficulties in diagnosing this FLL in clinical practice. Also, in all lesions it was possible to integrate elastography with US and CEUS, to obtain a multiparametric US approach easily applicable in practice. If we have an inconclusive result for the CEUS exam but a mean value of SWE showing a soft FLL compared to the surrounding parenchyma, having in mind the results of this study we could think of HCC.
We have added this information in the text.
page 6; line 272-280
Point 2: the key question is whether the technique distinguishes cancer from regenerative nodule. But, the study has just characterised HCC.
Response 2: Thank you for your comment. Yes, indeed, it is important to see if this method could differentiate HCC from regenerative nodules. Unfortunately, we could not analyze this due to the lack of cases with regenerative nodules and histological exams.
Point 3: The application may be limited in clinical practice as it is only useful once a focal lesion is identified on ultrasound to enable targeted measurement of SWE. However, the main problem with ultrasound is the fact that focal lesions are not seen/identified in the first place. If lesions are not recognized/identified on US, targeted SWE will not get measured in clinical practice.
Response 3: Thank you for your comment. We agree that visualization of a lesion in conventional ultrasound is essential and limits every ultrasound-based method for characterizing FLL, not only elastograhpy evaluation.
Point 4: Another limitation is that the study does not highlight how a low SWE reading would discriminate HCC from other hepatic focal lesion differentials.
Response 4: Thank you for your comment. We agree with the reviewer and we have highlighted this limitation of our study in the text. We had only a few cases of benign lesions in cirrhotic patients and that was the main reason for not including those cases in the study.
Point 5: The authors report finding significantly lower SWE readings for HCC in comparison to background cirrhotic livers. This could be helpful to improve the sensitivity of U/S in making HCC diagnosis. But, this hasn't been tested in this study.
Response 5: Thank you for your comment and for your suggestion. We were not able to make this analyse due to the limited number of cases with other FLL on liver cirrhosis.
Reviewer 3 Report
The Authors analyze elastographic features of hepatocellular carcinoma and the factors that influence intratumoral elastographic variability in patients with liver cirrhosis. Hepatocellular carcinoma may be diagnosed in diagnostic imaging then presenting certain enhancement pattern in the setting of chronic liver disease and/or cirrhosis. Based on the obtained results the Authors propose to include elastography in the multi parametric diagnostic algorithm of focal liver lesions. As such the paper fits into the special issue "Hepatocellular Carcinoma: The Current Recommendations for Clinical Practice”.
The papers generally reads well, however some minor language alterations are needed, e.g. line 52 - there is” are need” there should be „are needed”. Lines 54/55 - there is „ eventually avoid biopsy” , there should be „help to avoid biopsy” and further on.
I would also advise rephrasing the following sentence „This fact shortens the diagnostic algorithm duration, so that other potentially harmful and long-waiting procedures (such as CT/MRI and/or liver biopsy) can be avoided” (Lines 38-40). Especially in reference to further repetition, which is better phrased „Thus, 53 performing elastography during FLL evaluation can be a useful tool that can eventually 54 avoid biopsy (an invasive procedure) and resorting to CE-CT/CE-MRI (radiation expo- 55 sure and potentially nephrotoxic contrast agents, expensive, difficult re-attendance).” (Lines 53-56).
Again repetition: in lines 192-193 and 199-200.
The Authors are aware of the limitations of their study. Statistics seem appropriately applied. References are up to date.
Author Response
Dear Editor and Reviewers,
Thank you very much for your comments and suggestions for our manuscript JCM-1148663, entitled "Is there a place for elastograhpy in the diagnosis of hepatocellular carcinoma?", which are very valuable and have helped us revise and improve our manuscript. We appreciate all your work on our manuscript. We have carefully studied your comments and corrected our manuscript accordingly. We hope this version of the manuscript meets your approval. All changes have been highlighted in the manuscript. The corrections in the paper and the response to the editor and reviewers are as follows:
Response to Reviewer 3 Comments
Point 1: The papers generally reads well, however some minor language alterations are needed, e.g. line 52 - there is” are need” there should be „are needed”. Lines 54/55 - there is „ eventually avoid biopsy” , there should be „help to avoid biopsy” and further on.
Response 1: Thank you for your suggestions. We reviewed the English language and style and made the corrections accordingly.
Point 2: I would also advise rephrasing the following sentence „This fact shortens the diagnostic algorithm duration, so that other potentially harmful and long-waiting procedures (such as CT/MRI and/or liver biopsy) can be avoided” (Lines 38-40). Especially in reference to further repetition, which is better phrased „Thus, 53 performing elastography during FLL evaluation can be a useful tool that can eventually 54 avoid biopsy (an invasive procedure) and resorting to CE-CT/CE-MRI (radiation expo- 55 sure and potentially nephrotoxic contrast agents, expensive, difficult re-attendance).” (Lines 53-56).
Response 2: Thank you for your comment. We rephrased the sentence in order to avoid repetition.
Point 3: Again repetition: in lines 192-193 and 199-200.
Response 3: Thank you for your comment. We deleted the repetition.
Round 2
Reviewer 1 Report
Thank you for considering the comments and suggestions for this manuscript and made some revisions accordingly.
The main concern remains the clinical utility of the provided results. Regarding the value in clinical practice, the Authors replied that “if we have an inconclusive result for the CEUS exam of an FLL but a mean value of SWE showing a soft FLL compared to the surrounding parenchyma, HCC could be considered as a final diagnosis”. However, this reviewer does not agree with this point. For example, an atypical hemangioma can demonstrate an inconclusive enhancement pattern on CEUS and have a lower liver stiffness compared to the background cirrhotic parenchyma. Following the proposed approach and considering HCC diagnosis in lesions software than cirrhotic parenchyma, this could lead to a significant high false positive diagnosis and unnecessary treatments!! In clinical practice most of these lesions undergo biopsy rather than be considered as HCC.
Regarding the inter-reader agreement, the revised manuscript report that: “The ICC between the medians of the first five and last five EM was high and statistically significant (ICC: 0.926 (95% CI: 0.890-0.960, p=750)”. How can a p-value of 750 could be statistically significant? Usually, statistically significant values are p<0.05. In this case please report the exact p-value.
Author Response
Thank you for your comments and suggestions for our manuscript JCM-1148663, entitled "Is there a place for elastograhpy in the diagnosis of hepatocellular carcinoma?", which are very valuable and have helped us revise and improve our manuscript. We appreciate all your work on our manuscript. We have studied the new comments and tried to correct our manuscript accordingly. We hope this version of the manuscript is improved and meets your approval. All changes have been highlighted in the manuscript. The corrections in the paper and the response to the editor and reviewers are as follows:
Point 1: The main concern remains the clinical utility of the provided results. Regarding the value in clinical practice, the Authors replied that “if we have an inconclusive result for the CEUS exam of an FLL but a mean value of SWE showing a soft FLL compared to the surrounding parenchyma, HCC could be considered as a final diagnosis”. However, this reviewer does not agree with this point. For example, an atypical hemangioma can demonstrate an inconclusive enhancement pattern on CEUS and have a lower liver stiffness compared to the background cirrhotic parenchyma. Following the proposed approach and considering HCC diagnosis in lesions software than cirrhotic parenchyma, this could lead to a significant high false positive diagnosis and unnecessary treatments!! In clinical practice most of these lesions undergo biopsy rather than be considered as HCC.
Response 1: Thank you for your comment. It is a helpful observation that our message is not understood due to a vague statement. We agree with the reviewer, and we consider that elastograhpy is not enough for HCC final diagnosis when we have an inconclusive CEUS result, and further investigations are needed (other contrast-enhanced imaging methods CT/MRI, even liver biopsy). Through our study, we did not aim to underestimate the value of liver biopsy in HCC, especially in the era of personalized medicine where a biopsy can be mandatory for optimal treatment. We wanted to underline was that elastograhpy can be a helpful tool in orienting the diagnosis and the need for rapid further assessment. Keeping in mind that in the cirrhotic liver, hepatocellular carcinoma accounts for > 95% of all malignancies, in clinical practice knowing HCC stiffness could help orientate clinicians when having an inconclusive CEUS exam for a new FLL found in a cirrhotic patient.
The results of this present study show HCC elastographic features and propose further research for determining how best to incorporate these findings into clinical practice. Options could include software-based algorithms. Indeed, further studies are still required to obtain an evidence-based answer for the question raised in this paper's title.
We changed some statements and added new comments in the manuscript.
page 2; line58-63;
page 5; line238-242;
page 6; line299-308;
page7; line 339-440.
Point 2: Regarding the inter-reader agreement, the revised manuscript report that: “The ICC between the medians of the first five and last five EM was high and statistically significant (ICC: 0.926 (95% CI: 0.890-0.960, p=750)”. How can a p-value of 750 could be statistically significant? Usually, statistically significant values are p<0.05. In this case please report the exact p-value.
Response 2: Thank you for your comment. We meant that the ICC (intraclass correlation coefficient) between the medians of the first five and last five EM is high and statistically significant (ICC: 0.926). p values indicate the significance of the Friedman test: comparison of the first five versus the last five elastographic measurements that show no difference between the first and last five measurements, and the correct spelling is p=0.75). We clarified this data in the manuscript.
page 5; line216-228.
Reviewer 2 Report
Authors have modified the manuscript according to the suggestions made.
Author Response
Thank you for your review. We appreciate all your work on revising our manuscript.